# Study on DID Application Methods for Blockchain-Based Traffic Forensic Data

Cheolhee Yoon [1] , Jaehun Hwang [1], Minje Cho [1] and Bong Gyou Lee [2,*]

1 Police Science Institute, Korean Nation Police University, Asan 31539, Korea; bertter@police.ac.kr (C.Y.); sharproo@police.go.kr (J.H.); epiworld@police.go.kr (M.C.)
2 Graduate School of Information, Yonsei University, Seoul 03722, Korea
* Correspondence: bglee@yonsei.ac.kr; Tel.: +82-10-4746-3090

**Abstract:** Traffic accident investigation has been used to reconstruct the cause and condition of the accident by using images and driving record data that introduce IT technology, and to analyze both sides of the accident, skid marks, and vehicle damage to identify the perpetrators and victims. However, level 3 self-driving vehicles are the most important factor in determining the cause and imputation of the accident by the driver or manufacturer with control information at the time of the accident. It is also developing into a network and connected vehicle with various restrictions such as the burden of the price of sensors equipped with self-driving cars and climate and rapidly changing road traffic information. In addition, network and connected vehicle driving data are stored on the outside, or various devices and sensors are installed to store information on the outside for convenience in operation, and efforts to enact laws are continuing. This paper attempts to propose a traffic investigation digital framework using digital data generated by these devices and sensors.

**Keywords:** vehicle forensics; digital forensics; infotainment; traffic accident investigation; DID; self-sovereign identity





## 1. Introduction

When a traffic accident is reported to the police station in the vicinity, the on-site officer and the traffic accident investigator identify the cause of the accident based on various types of evidence (e.g., skid marks, a vehicle's damaged parts, simulation etc.) and identify the perpetrator and the victim, with many refusing to acknowledge that they themselves are the perpetrator. Recent traffic accident investigation involves the use of video records and driving records (EDR, DTG) to identify the cause of the accident and distinguish the perpetrator from the victim. When there are no video records or driving records, however, there are difficulties in determining the exact cause of the accident when the investigation is based on statements from both parties.

Self-driving cars should be designed and developed considering a variety of cases, including vehicle structure, various sensors, actuators, communication failures, potential software errors, inadequate control, potential collision possibilities, off-road, loss of stability, violations of traffic laws, and abnormal driving. Among the matters considered in the design of self-driving cars, the Ministry of Land, Infrastructure, and Transport recommends that mode data should be traceable in the final report of the 2018 self-driving car convergence future forum. In addition, the early self-driving vehicle was equipped with sensors in the vehicle to recognize the surrounding environment and drive autonomously based on the collected information. However, to overcome various constraints such as the price burden of sensors and limitations of perception by weather or road environment, we are developing them by sharing various information through communication with roads and infrastructure. In advanced countries such as the U.S., Germany, and the U.K., Cooperative ITS (C-ITS), which focuses on the exchange of information between combined

vehicles and road infrastructure, is actively promoting in advanced countries such as the U.S. and Europe [1].

Referring to Figure 1, Digital data collected by sensors for autonomous driving is stored within a vehicle, but numerous IT companies and motor companies such as Samsung, Harman, Microsoft, and Baidu are developing ways to store data in a cloud because of the large amount of data. South Korea, in particular, defined the method of transmitting and storing vehicle operation data to the center in real time, as in the figure below, based on location (track) information for safe driving and operation status information based on the location of the RSU by way of wireless communication. When a vehicle equipped with telematics service transmits CAN or ECU information, vehicle status information (direction, speed etc.) obtained through OBD-II, location (track) information calibrated by correcting the GPS received information within the vehicle, and information on vehicle events to a nearby roadside device (RSU), the location-based vehicle data for each vehicle collected by the roadside device is automatically sent to a center at regular intervals or when it occurs and stored in a database (DB) within [2].

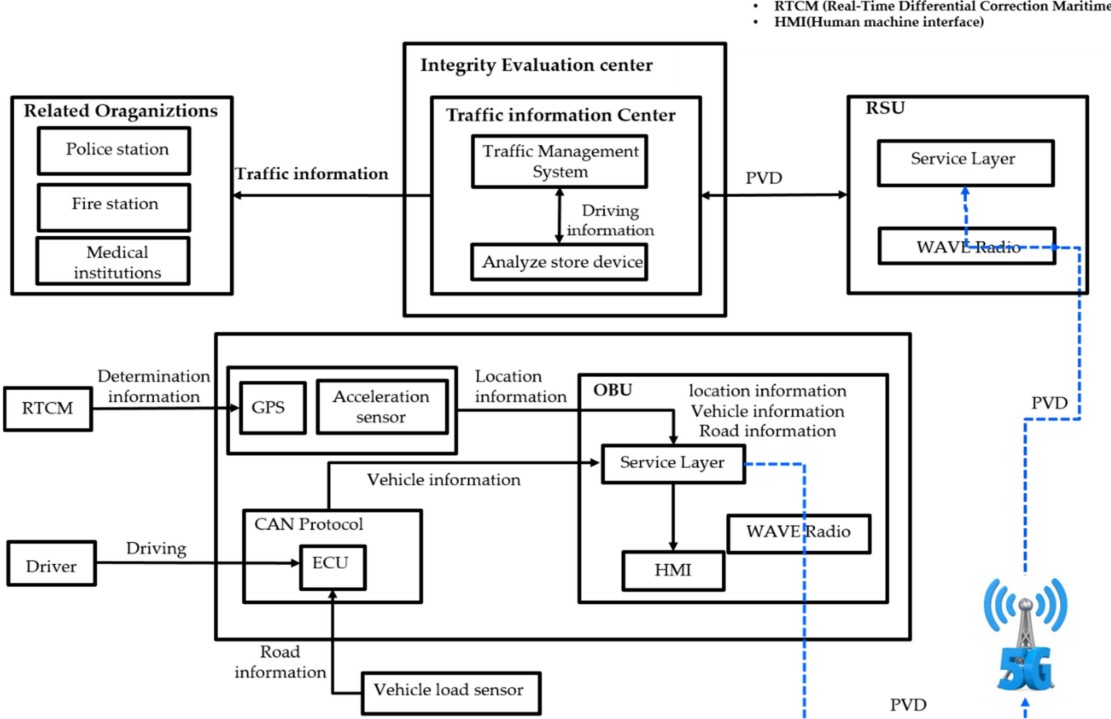

**Figure 1.** Location-Based Vehicle Data Collection Service.

Countries leading the development of self-driving cars such as the U.S., the U.K., Germany, and Japan are pushing for laws to identify the attribution of responsibility for accidents in Level 3 self-driving vehicles from drivers to manufacturers or software developers [1]. However, there is no way to prove the cause of the accident. Even if the law is enacted, the law is useless. Therefore, in order to prove the cause and responsibility of the accident, the integrity of the accident data must be ensured, among other things, for accuracy and fairness. Therefore, using blockchain technology, a technology that has proven the accuracy and integrity of current data, it is possible to identify the exact cause of the accident.

Section 2 identifies how the vehicle communicates or types of data for driving, Section 3 identifies digital data that can be obtained from the vehicle, and Section 4 describes the distribution of blockchain-based vehicle accident data. Existing papers emphasized data distribution and integrity through blockchain, but our paper emphasizes the distribution of vehicle accident data through self-main identity technology. The distribution

of metadata at the time of the blockchain-based accident will be very helpful in resolving the vehicle accident, as it is important whether or not there is an accident of autonomous vehicles at Level 4 in the accident or not.

## 2. Materials and Protocols

A map, which is a part of finding the optimal route for a vehicle and driving, is divided into three types of maps; a navigation map, an ADAS map, and a precision map (LDM: Local Dynamic Map). First, the navigation map that is used on vehicles or cell phones considers traffic conditions and displays the optimal route when a destination is stated. An ADAS map, which is more advanced than the conventional navigation map, assists in making the ride smoother, for it contains information on the curvature and inclination of roads. However, LDM, which is the most essential in autonomous driving, goes beyond the conventional recognition of roads and contains all information about roads such as lanes, centerlines, traffic signals, and guardrails down to an accuracy of within 10 cm. The existing navigation system or an ADAS map is restricted in times of rain or snow, but an LDM contains all the information about roads and can provide the optimal driving conditions without restrictions such as the weather. A CAN (Controller Area Network) is a standard protocol for the communication of numerous electronic control units (ECU). A CAN is widely used in vehicle communication for its universality and efficiency, but issues regarding improvement of its functions and transmission bandwidth have been raised due to the increased number of ECUs installed in a vehicle and the addition of advanced sensors. As such, various protocols such as LIN, FlexyRay, MOST, and CAN-FD were suggested, but none have been able to replace the universality of CAN and are only used for specific parts or OEM [3]. An Automotive Ethernet is also being considered as a new type of vehicle communication network-based technology that accommodates other networks and supports the existing TCP/IP system as well as a wider bandwidth for the increased functionality of CAN. When a so-called Connected Car is connected to a network (e.g., 5G, DSRC etc.) and autonomous driving and connected technology is applied, it is very likely that communication between each ECU and Actuator will be processed by an Automotive Ethernet [3].

CAN uses the OBD-II port to determine whether vehicle sensors have failed and uses Diagnostic CAN protocol to diagnose a vehicle. An OBD-II port is also used to obtain information from a vehicle's dashcam or navigation system and as an HUD (Head-Up Display) for driver convenience. Only some of the information gathered from sensors defined by IBD-II, such as RPM, vehicle speed, throttle position, information on fuel, and steering, may be used in traffic accident investigations. There are two types of devices that record a vehicle's operation information: An EDR (Event Data Record) and a DTG (Digital Tacho-Graph). Analyzing the data record in an EDR can determine impact velocity, the use of brakes or accelerator, and whether a seatbelt was fastened. A DTG is required by law on commercial vehicles such as taxis, buses, and trucks since 2013 and is proving very useful in traffic accident investigations [4,5]. Traffic accident analysis using EDR or DTG is more detailed and reliable in analyzing the vehicle that is the subject of a traffic accident compared to the traffic accident analysis method of analyzing damage to a vehicle or skid marks, and has great value as forensic evidence [6]. It is usually difficult to adopt video records used for analyzing traffic accidents as evidence for accident analysis because analyzing whether the driver is at fault, the issue of picture resolution, altering, falsifying, or damaging records are common issues. Digital data generated by various sensors attached to a vehicle is not easy to alter or falsify and are stored in log form, making it valuable as evidence and also making it possible to introduce it to accident investigation or a digital forensics framework. Since there is no framework that applies digital data generated by sensors, however, we would like to propose introducing a framework through case studies of accidents utilizing digital data.

### 3. Forensic of Digital Data Collected from the Accident Vehicle

*3.1. Forensic of ECU(Electronic Control Unit) Vehicle Sensor, ADAS Vehicle Sensor, and LIDAR*

Per Diverse technology is being applied to vehicles along with advancements in IT. As vehicles become more advanced, the use of ECUs (Electronic Control Unit) is also on the rise. An ECU has control over various aspects like playing a key role in driving involving the engine, brakes, and steering system as well as devices for user convenience such as air bags, the audio system, and the air conditioner. As vehicles become more advanced, more ECUs are being installed in them. Although there are some differences according to vehicle type and specifications, electric parts can be divided into the powertrain, chassis, body, infotainment, safety, and security parts. The powertrain is in charge of engine control, the self-diagnosis device, and cruise control, the chassis is in charge of steering, brakes, and suspension, while infotainment is in charge of voice recognition, the navigation system, and multimedia. The safety and security parts include self-parking, anti-collision, radio remote locking, and tire air pressure. Data on vehicle speed, acceleration, and deceleration status, wearing or not wearing a seatbelt, and airbag deployment before and after the accident is collected from vehicle sensors and stored in the EDR. Vehicle sensor data usually occurs in real time during operation of a vehicle, so it is expected to be stored on an external server through an external storage device or communications. A method of determining curved lanes by combining a direction sensor and an acceleration sensor was presented [7]. Two sensors were attached to a vehicle, which was driven on a road, and the data obtained was used to detect straight lanes and curved lanes. In this study, curved roads were not detected when driving at nighttime or in the rain and curved lanes were not detected when they were disconnected, which were existing limitations. However, it was presented that analysis is possible without being affected by the environment when using both sensors. Research and development of a system for storing data generated during driving on high-speed imaging-based smart cameras was conducted. Using inter-vehicle communication functions, we have announced how to collect and store video records obtained from cameras included in the device, RPM, vehicle speed, information on the positions of the acceleration pedal, the gear, the steering wheel, and the deceleration pedal [8]. And ADAS (Advanced Driver Assistance System) is a system that uses various advanced sensors, GPS, communications and intelligent imaging equipment to enable a vehicle itself to perceive its surroundings, assess the situation, and then control the vehicle or use warning sounds, lights, or vibrations to alert the driver of risk factors, thereby helping a driver to drive safely. Vehicles recently released by manufacturers (e.g., Mercedes, Aud, Hyundai etc.) which are equipped with high tech functions and most of the functions enabling safe driving are ADAS technology [9]. With the development of ADAS technology and when connected to a network (like 5G or DSRC) to conduct V2X (Vehicle to everything) service, which means object communication, Level 4 automotive driving will be possible by using information on surrounding road conditions (e.g., nearby vehicles, ADAS maps, precision maps etc.) as s Visualization of ADAS sensor information.

Sensors used in ADAS use Inertial Measurement Unit (IMU), Global Positioning System (GPS), cameras, and LIDAR for location recognition, while cameras, LIDAR, and RADAR are used for identifying objects. Cameras enable accurate identification of form information, including traffic sign recognition, blind spot detection, and lane departure. RADAR (RAdio Detection And Ranging) sends out electromagnetic waves into the air and uses the waves that bounce off the objects to detect their direction and speed. Because laser uses microwaves, it can measure distance with stability night and day and complements the camera. LiDAR (Light Detection And Ranging) works on the same principle as RADAR but uses high-powered pulse laser so the distance information it gains is different. It sends out millions of laser beams in one second to measure distance, so it combines all the collected information and can reconstruct it in 3D. RADAR and image sensors create approximately 100 MB of data per second, and the internal ECU creates 50 MB of fusion sensor data per second. For example, the data collected by 5 RADAR sensors and 2 image cameras per second is estimated to be about 1 GB. If an autonomous driving vehicle were to drive for

one day with a generated data rate of 1 GB per second, the total amount of data needed would be approximately 30 TB [10]. Advances in ADAS technology are made with the goal of fully autonomous driving, and since the amount of data collected by cameras and various sensors for fully autonomous driving is increasing, it would be difficult to store it within the vehicle. Therefore, Samsung Electronics-Harman is developing autonomous driving technology based on cloud serving without a computer within the vehicle while Microsoft Corp. is developing a cloud and AI-based autonomous driving service (e.g., Volkswagen Automotive Cloud, Baidu's Apollo Alliance etc.). As such, there is a tendency to shift to external storage without storing driving data internally. Accordingly, the collection of vehicle data for vehicle digital forensics also needs to change to a method of analysis that utilizes external data.

### 3.2. Forensic of Digital Sensor Data Stored in Infotainment and Log Data

A vehicle's infotainment system (e.g., navigation system, TPEG, assisted parking etc.,) supports driving convenience (e.g., OBD, ADAS, communication between vehicle and the infrastructure, communication between vehicles), safe driving, and infotainment (e.g., DMB, audio system, games, mobile office etc.). Table 1 shows The table below lists the basic infotainment functions of a Ford Fiesta, which does not have a navigation system. It is possible to see the information used by the user, such as the call record, the information of the connected device, information on texts, and Internet use, and the information stored in the system [11].

**Table 1.** Forensic data of infotainment.

| Data | Type | Location | Meaning |
|---|---|---|---|
| Device Lists | User Data | MediaCache/ | Names |
| Device Serial Number | User Data | MediaCache/ | |
| Device playlist | User Data | MediaCache/ | |
| Contact Names | User Data | GrammarFSM/ Windows/phonebook iVe Report | Arbitrary files |
| SMS | User Data | TxtMsgApp/ | Potential SMS information |
| Registry | System | Documents and Setting | |
| User Activity | User Data | Windows/LogFiles/ | |
| Windows Dump | System | Windows/LogFiles/ | Windows memory dump |
| System Events | System | Windows/ LogFiles/ | |
| Internet History | Application | Cookies/ Temporary | |

VMDS (Vehicle Mounted Data System) monitors vehicle information in real time and also analyzes and stores information pertaining to the vehicle and the driver. This system obtains vehicle and location information from the terminal installed in the vehicle at regular intervals and then stores and analyzes it on the control server and keeps information on driving characteristics, operation statistics, and operation history. Also, it provides real time vehicle diagnostic information, making it easy for the driver or vehicle manager to respond in the event a failure occurs, and an operation log is created automatically to improve bad driving habits and reduce vehicle maintenance costs [12]. The information collected for vehicle information analysis includes vehicle speed, RPM, mileage, and vehicle location information using fuel injection time and GPS, which is transmitted to the server in real time. By utilizing time and driving information related to an accident like GPS information, which is data collected in the VDMS, vehicle speed, and RPM, it is possible to determine

the speed at collision and whether speed was reduced when an accident occurs to identify the possibility of avoiding the accident. Probe Data Service on Location is closely linked to autonomous driving vehicles. The collection of Probe Data on Location is based on WAVE Communication and is a service where all vehicles and roadside base stations collect location (track) information and driving status in real time from the device installed on the vehicle and transmit it to the traffic control center where it is stored. When Probe Data Service on Location is supported, collection of driving records will become much faster and investigation using more objective data is expected. As shown in [13,14], it is possible to identify the location of the accident by using the collected information, and more accurate accident investigation is possible through the use of driving-related information such as speed at the time of the accident and RPM.

## 4. Digital Forensic Framework by Application of Blockchain-Based Traffic Accident

*4.1. Vehicle Sensor Analysis Based on Traffic Accident Digital Forensic Framework*

The application of blockchain technology in this area can be prevented from traffic accidents. Because we use smart contracts could enable auction mechanisms. Because we use smart contracts, we can easily find our destination and avoid congestion, so we can prevent accidents in advance [15,16].

As shown in Figure 2, the main goal of smart cities is to improve the services offered to citizens, reducing administrative costs through the use of technology. Considering that the use of IoT, although very useful, leads to security problems in data management, the introduction of blockchain technology could solve many important issue. Privacy and security [17]. In other words, implementing blockchain technology could mean making drivers more secure in sharing information about real-time position, traffic conditions, and any unexpected events, such as road accidents. The trust management problem in vehicular networks could be solved by blockchain technologies, and new, secure systems based on blockchain technology could support [18].

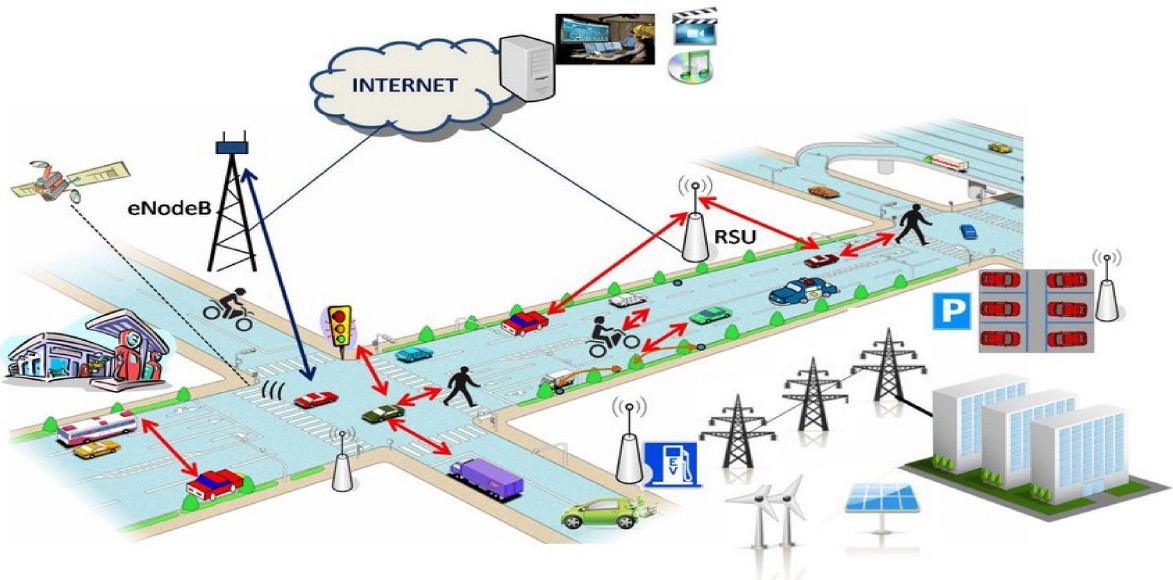

**Figure 2.** Conceptual Diagram of Autonomous-Driving Data Infrastructure.

The collected data showed time (the time the vehicle was driving), latitude, longitude, heading (the value of overcoming the limitations of the accuracy of latitude and longitude), and speed. This analysis wasn't quite complete as the vehicle was only driven straight, and no information on the steering wheel angle was collected. When an accident investigation is conducted based on the information collected, however, it is possible to identify the speed at the time of the accident by using time and speed and identify the location of the traffic accident by using latitude and longitude. Also, by checking the change in speed in

relation to the time of the accident, it is also possible to analyze that there was an attempt to avoid the accident if there is a sudden decrease in speed or that the driver made no attempt to avoid the accident if the speed continued at a consistent rate.

The accident investigation of self-driving cars should be conducted based on various driving-related information, such as whether the information delivered to the sensors inside the vehicle, whether there are any defects in driving software, and whether the device properly ordered the information received from outside, not only by the driver's fault.

And the SAE J2735 uses standard Probe Vehicle Data (PVD) type Full Position Vector and Local Content fields. The meaning of each field is as Table 2.

**Table 2.** Probe Vehicle Data.

| Name | Details |
|---|---|
| DSRC msgID | Unique identifier which signifies upload of collected information (PVD: 10) |
| DDate Time | GPS time information (in accordance with SAE J2735 standards) |
| Longitude | Longitude location information (in accordance with SAE J2735 standards) |
| Latitude | Latitude location information (in accordance with SAE J2735 standards) |
| OBE ID | Unique identifier assigned to OBE |
| Manufacturer/Model | OBE manufacturer and model name |
| Firmware version | Firmware version information |
| Firmware Upgrade Status | Firmware download or upgrade status |
| Map version | Map version |
| RSSI | Strength of signal reception |
| Traffic Collection Information | Traffic information collected by OBE |

In particular, existing accident investigations were subject to punishment by drivers. However, self-driving cars can be targeted by manufacturers, operating system developers, and driving assistants. Therefore, a variety of vehicle driving information should be collected that may be the cause of the accident. However, the collected data must be analyzed in a state of integrity without forgery modulation to be valuable as evidence.

### 4.2. Digital Forensic Framework of Vehicle Sensor Analysis Based on Blockchain

Various kinds of sensor data from a vehicle accident collected for traffic accident investigation must be provided to multiple verification authorities without being hacked or altered for evidence analysis use. The vehicle accident sensing data to be provided to multiple specialized agencies can use DID applied traffic accident digital forensic framework for managing a blockchain-based self-sovereignty identity. The transparency and integrity of the video records and driving records stored at the time of the accident is guaranteed by a blockchain-based platform, and this also remains as important evidence for determining the cause of a traffic accident following an investigation through a traffic accident analysis digital forensics platform and a procedural analysis process. The blockchain generally utilizes an intelligent contract to automate contract terms, a hashing algorithm to safeguard information confidentiality, a consensus mechanism to safeguard data integrity, and an asymmetric key to safeguard data flow security [19].

The DID(Decentralized Identity) applied for managing self-sovereignty identity manages the contents of personal information and various kinds of verification directly or verifies and manages them by sharing each issuing agency's public key based on trust as seen in Figure 3. The components for managing the self-sovereignty identity if vehicle

accident sensor data consist of DID and DID documents used as an identifier and a way of verification, VC (verifiable Credential) used as an ID for storage, and VP(Verifiable Presentation), which is used as an ID for submission. The main participants of an investigation of accident vehicle sensor data are the Issuer, which issues the VC of the major participants, the Holder, which processes the VC it receives into VP and submits it to the verifying agency, the Verifier, which receives the VP from the Holder and verifies its authenticity, and the Identifier Registry, which is a distributed repository that stores DID and ID-related information. VC is authentication of vehicle identification such as the vehicle registration card that was issued to the vehicle user by the issuing agency and the vehicle identification number. However, the user does not use the VC directly and processes it into VP which is ID for submission, and applies vehicle accident data DID. From the properties of the resident registration card VC of the vehicle owner and the vehicle registration card VC, for example, the name and age are extracted from the resident registration card VC while the vehicle identification number is extracted from the vehicle registration card, which are then created into a single VP and consequently submitted to the agency investigating the accident. The DID in this case is used as the identifier that goes in the VC and the VP, so it is used as the verifying identifier for securing the integrity of the accident vehicle sensing data [20]. The DID (Decentralized Identifier) infrastructure is realized on a DID compatible blockchain, distributed ledger, or distributed network data platform. Data collected at the vehicle accident site should also be stored in a database on a blockchain platform for the purpose of collecting evidence, and the collected data should convert evidence information to DID and manage it by using a public key, verification protocol, and service end pointer data set which are needed for interaction between encrypted entities. This can later be carried out as digital forensic activities which are applied to distribution and investigation for digital forensics analysis investigation. In the case of DID as seen in Figure 3, the guarantee of integrity and authentication can be processed by the agency itself through the public key shared by agencies connected to a dispersed network, and integrity can be guaranteed in the process through a consensus algorithm [20].

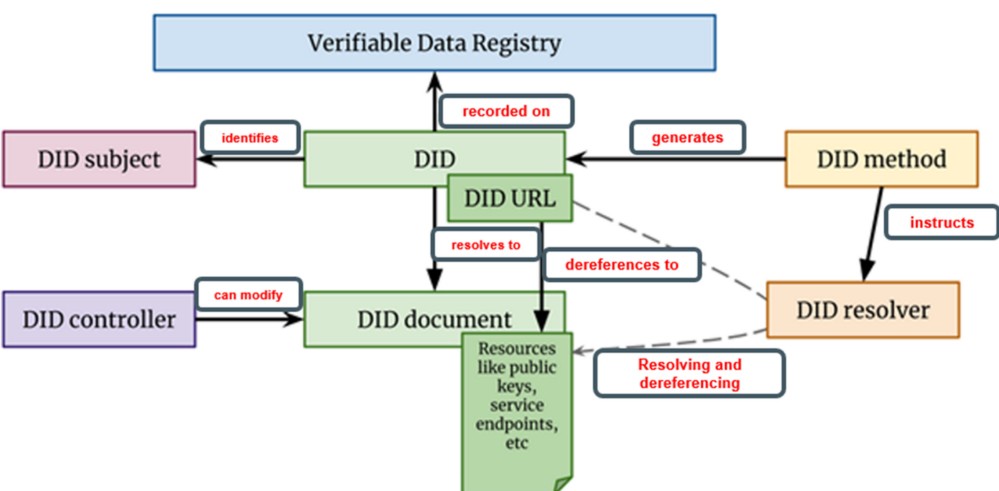

**Figure 3.** Data Decentralized Identifier (DID) Distribution on Blockchain.

Traffic accident data analysis requires data before and after an accident to determine the cause of the accident, and data should be stored before and after the time of transition of vehicle control or system failure for accident prediction analysis of the accident. This data can be compared and analyzed through similar historical situation data for comparative analysis of event data such as accidents, transfer of driving control, and system errors. In addition, since the vehicle is stored in binary form for real-time storage of large data, the source data of the self-driving vehicle is essentially separated via Post-processing and subsequently enhanced by its own systems, such as DID-based cloud servers, to store it in a standardized index DB format [21].

### 4.3. Digital Forensic Framework Solution of DID on Blockchain

Figure 4 shows the data generated from the vehicle's sensor by applying DID. Verifiable Credential through a Verifiable Data Registry contains important verification information such as information on the accident vehicle, the issuer, qualifications, and the issuer's signature, so additional management is conducted in encrypted form or a wallet. And we need to be clear that blockchain and DIDs are not used for large amounts of data. Instead, they are suitable for small amounts of important and valuable data, including metadata [19,21]. First, when a DID document is registered in a blockchain type distributed content repository after DID is created by the vehicle user and each related agency including the police. Because DID is not necessarily created by the vehicle user, It depends on which DID such as DIDs come from the auto manufacturer, come from the auto insurance company or some other institution. The vehicle's DID and secret key are stored safely through an app installed in the victim or suspect's mobile terminal, while each agency's DID information and the secret key is managed within the server that each agency operates. Then the unique information of a vehicle's accident data sensors can be utilized, and also digital forensic analysis is possible with integrity, guaranteed through the DID of expert agencies. Certification is made in the order shown in the following table.

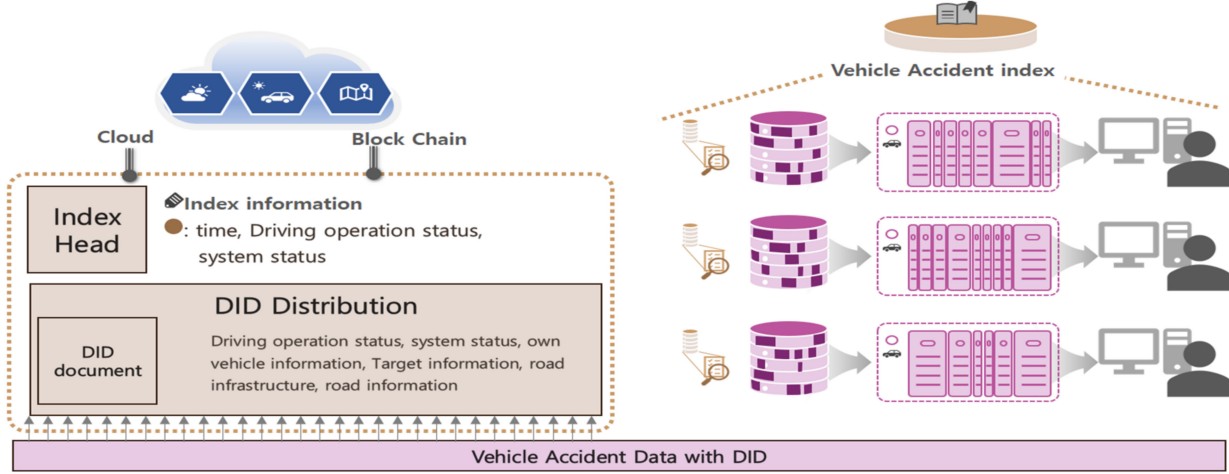

**Figure 4.** Vehicle Accident Data DID Distribution on Blockchain.

Create a key Pair, prepare credential for issuing, issue verifiable presentation, verifiable presetition.

In addition, if the attached accident vehicle data is inserted into the blockchain by applying a hash and then distributed, integrity is guaranteed. First, data created in the accident vehicle from all the sensor data was verified. The sources were the accident vehicle and the receiver, which detects the destination and the area the vehicle passes through. When this is verified, the framework for traffic accident forensics is guaranteed, such as Table 3. Moreover, the estimate data for accident analysis is generated as shown in Table 4 below, and the meta values of sensors, vehicles, and control data are loaded into DID Document and distributed [17,22,23].

**Table 3.** Certification of vehicle accident data.

| Create a Key-Pair If Needed | Get DID (Did:Key) for Key-Pair |
|---|---|
| if [ -e key.jwk];<br>then<br>    echo 'Using existing keypair.'<br>else<br>    didkit generate-ed25519-key > key.jwk<br>    echo 'Generated keypair.' | did=$(didkit key-to-did-key -k key.jwk)<br>printf 'DID: %s\n\n' "$did" |
| **Prepare credential for issuing** | **Issue verifiable presentation** |
| {"@context": "https://www.w3.org/2018/credentials/v1",<br>    "id": "http://example.org/credentials/3731",<br>    "type": ["VerifiableCredential"],<br>    "issuer": "$did",<br>    "issuanceDate": "2020-08-19T21:41:50Z",<br>    "credentialSubject": {<br>        "id": "did:example:d23dd687a7dc6787646f2eb98d0"<br>    }<br>} | didkit vc-issue-presentation \\<br>    -k key.jwk \\<br>    -v "$did" \\<br>    -p authentication \\<br>    < presentation-unsigned.jsonld \\<br>    > presentation-signed.jsonld<br>echo 'Issued verifiable presentation:'<br>print_json presentation-signed.jsonld<br>echo |

**Table 4.** Certification of vehicle accident data.

| Condition | Data |
|---|---|
| Sensor | LIDAR Data, (3D Point Cloud), Lane Detection Results (MobileEye), RADAR Object |
| Vehicle | IMU (Inertial Measurement Unit), GPS (Global Positioning System), Vehicle Location Information |
| Drive Support Data | LDM (Local Dynamic Map)<br>Real-Time driving data (leading Car, Vehicles driving in Front, Cut in) |
| Driving Strategy Data | Vehicles driving in Front, Flag/Speed Bump Flag/Cut-in Flag/Crosswalk stop Flag/Crossroad Flag/Traffic Light stop Flag (Include Stopping distance)/Left turn at your own risk Flag/Stop at Rotational Intersection Flag(Include stopping distance) |
| Control data | Curvaturn information, Vehicle drive Estimated Path (Quadratic curve Function),<br>Control: Steering, Acceleration or Deceleration |

## 5. Conclusions

Recent traffic accident investigations analyze the video records and driving records installed in a vehicle and use them as important evidence in identifying the cause of an accident. When there is only a video record or only a driving record or neither is available, there are many instances when the cause of the accident is undetermined because investigation relies solely on the statements of the perpetrator and the victim.

This paper proposes and applies a traffic accident investigation framework that utilizes the digital data created by sensors and devices installed in a vehicle even when the vehicle has no video record or driving record. In the case of video analysis, there was difficulty in analysis because drivers would damage the data, or recording was not done properly, or the driver refused to submit it when it was disadvantageous to them. Difficulty in analysis caused by the picture quality of the video information can be overcome by using the unique information that sensors have and also conducting digital forensic analysis with data integrity through the DID of professional agencies. Information from sensors on future autonomous driving vehicles will be impossible to forge or alter from a digital forensic perspective and will be stored in log form, so it has high value in that a professional agency will need to commission to analyze it.

The problem with driving records of a vehicle involved in an accident to date is that they cannot be collected in real time at the scene of the accident. It requires a tremendous amount of time to collect evidence pertaining to an accident, which causes difficulty in solving a case. If a DID-applied digital forensics platform is used, it is possible to collect

sensors and various other digital media in real time and also display the time, speed, and distance traveled on the map to be used in traffic accident investigation. The digital data created by sensors can identify a variety of causes for an accident that cannot be analyzed on video, such as speed at a particular time, a vehicle's location information using sensors, steering wheel values, and the routes the vehicle traveled. Applying DID (Decentralized Identity) for managing the identity of a blockchain-based self-sovereignty guarantees the execution of the digital data forensics of such sensors. It is time for autonomous driving vehicles to start operating at Level 3 and for preparations to be made for related vehicle digital forensics. At the same time, research on traffic accident sensor data digital forensics for digital forensic accident investigation on vehicles, including autonomous driving vehicles, should also be conducted. In particular, data generated from vehicles can actively perform digital data forensics by applying DID based on blockchain.

**Author Contributions:** Conceptualization, J.H. and C.Y.; methodology, M.C.; software, C.Y.; investigation, B.G.L. and J.H., writing-review and editing, C.Y. funding acuquistion, C.Y. All authors have read and agreed to the published version of the manuscript.

**Funding:** This work was supported by Institute of Information & communications Technology Planning & Evaluation (IITP) grant funded by the Korea government (MSIT) (No.2020-0-00901, Information tracking technology related with cyber crime activity including illegal virtual asset transaction).

**Data Availability Statement:** Software, python code and block chain DID in future sense platform; Data, Vehicle sensor data.

**Conflicts of Interest:** The authors declare no conflict of interest.

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
