# Peer review of "Study on DID Application Methods for Blockchain-Based Traffic Forensic Data"

_applsci, doi:10.3390/app11031268_

Round 1

Reviewer 1 Report

The authors are suggested to review the following paper and discuss data integration of various car developed by different manufactures using different protocols

Seljeseth, M., Yamin, M. M., & Katt, B. (2020). UIOT-FMT: A Universal format for collection and aggregation of data from smart devices. Sensors20(22), 6662.

Moreover, the authors are also encouraged to review literature from passive traffic sensors and information correlation through them that are being used for traffic management 

Finally, the authors are requested to improve the formatting of tables and use the listing for JSON as it is not suitable for the journal publication

Author Response

We Corrected opinions related to data integration. The table has been modified in the MDPI common format.

Reviewer 2 Report

The authors claim to present a traffic digital framework using the digital data generated by devices and sensors.

I suggest to reject this paper. My reasons are explained below.

  1. First of all the contribution of the authors is unclear. The paper is classified as a "Review", but the authors do not use a rigorous procedure to collect the most important references about the topic. A well-structured Review should have a collection phase, an analysis phase, and should address a set of future challenges about a certain topic. But, none of this is present within the paper.
  2. Many acronyms are not defined before they are used, so understanding the article is really difficult.
  3. The authors claim that "Since there is no framework that applies digital data generated by sensors, however, we would like to propose introducing a framework through case studies of accidents utilizing digital data." However, nowadays, there many papers about the application of the blockchain technology for traffic/accident management. In most of these, a framework is proposed, but they are not cited. I think that more references about the application of the blockchain technology for traffic/accident management are necessary to improve the overall quality of this paper. 

  4. I think the paper should be revised by an English Native Speaker because there are many mistakes.
  5. The writing of the paper is not fluid and there are several repetitions and errors (for example you refer to Figure 14 and it is not within the text).

I can not see a real contribution by the authors, and above all, this paper in my opinion, cannot be classified in the current status as a review in the scientific sense of the term.

Author Response

Explained the acronym..In addition, we refer to the block chain technology related data for traffic accident management.

Reviewer 3 Report

Authors of the work “Study on DID Application Methods for Blockchain Based Traffic Forensic Data” proposes and applies in this paper a traffic accident investigation framework that utilizes the digital data created by sensors and devices installed in a vehicle even when the vehicle has no video record or driving record.

Overall, it is a well-structured paper; the introduction section is wide and presents the purpose of the research in detail. There is a comparison and evaluations of the proposed method, using proper figures and showing up the proofs of the experiments included in the paper.

Although the proposal is interesting and within the scope of Journal of Applied Sciences, there are different issues that should be addressed in order to improve the work.

[Minor Comments]

  • Improve the state of the art about Blockchain and their use in this kind of systems focus on traffic forensic data.
  • There are multiple Acronyms (EDR, DTG, RTCM, HMI, etc), which must be explained and detailed.
  • Section 4.3 should be more detailed about the Certification of vehicle accident data.
  • It is recommended to compare the proposed filter with other proposals and fire detection systems that can help to complement the state of the art as (some examples that can help but you can find more). And even improve some aspects of the case study like the presentation of all the parameters and detection algorithms:
    • Mezquita, Y., Casado-Vara, R., GonzÁlez Briones, A., Prieto, J., & Corchado, J. M. (2020). Blockchain-based architecture for the control of logistics activities: Pharmaceutical utilities case study. Logic Journal of the IGPL.
    • Cebe, M., Erdin, E., Akkaya, K., Aksu, H., & Uluagac, S. (2018). Block4forensic: An integrated lightweight blockchain framework for forensics applications of connected vehicles. IEEE Communications Magazine, 56(10), 50-57.
    • Valdeolmillos, D., Mezquita, Y., González-Briones, A., Prieto, J., & Corchado, J. M. (2019, June). Blockchain Technology: A Review of the Current Challenges of Cryptocurrency. In International Congress on Blockchain and Applications (pp. 153-160). Springer, Cham. Updating with more recent references (Within the last 5 years).
  • Updating with more recent references (Within the last 5 years).
  • The article's English must be reviewed by a native speaker.

Author Response

We have added a description of vehicle accident data authentication through DID. Explained the acronym Also we updated to the latest reference.

Reviewer 4 Report

The following are my conclusions:

- the paper describes an interesting topic;
- the contributions of the paper are based on realistic and referenced assumptions;
- the problem in the manuscript is well defined, and the objectives are clear;
- the paper adequately put the progress it reports in the context of previous works, representative referencing and introductory discussion
- the conclusions and potential impacts of the paper are made clear.
- it can be possible enrich the bibliography with other paper (e.g. A survey on driverless vehicles: from their diffusion to security features” or “An overview of big data analysis”).

Author Response

We updated to the latest reference.

Round 2

Reviewer 1 Report

The research work can still be improved by adding further analysis in section 4 or by adding a new section 5 for comparing the performance of existing technologies with the proposed framework 

Author Response

Reviewer

We thank the reviewer for this insightful comment. In responses, the text of the manuscripts was modified in the following manner( page 9, line 15-20).  And I appreciated the constructive criticisms of distribute of vehicle data by Did with block-chain. We realized that the initial text may have been unclear, So we changed the section 4,5.  We checked more papers,  “Blockchain and Smart Contracts for the Internet of Things”, “BlockChain: A Distributed Solution to Automotive Security and Privacy”, “ An integrated lightweight blockchain framework for forensics applications of connected vehicles”

The entire manuscript has been carefully edited. As a result, the clarity and readability of the manuscript have been improved.

We look forward to hearing from you in due time regarding our submission and are happy to respond to any further questions and comments you may have.

Reviewer 2 Report

The quality of the paper has only partially improved compared to the previous version and the authors have not provided point-to-point responses to my observations.

Please, send a list of point-to-point responses about the following aspects:

  1. I would like to understand if the paper, according to the authors, is a "Review" or not. I can not see a standardized and well-definite procedure for classifying the paper present in the literature about the topic. 
  2. The references should be numbered, following the order of appearance within the paper.
  3. The literature review about the use of the blockchain-technology in transportation is very very limited. I think that you should cite the papers listed below, in order to improve the quality of the manuscript and to make your contribution clearer.  [R1] gives a very good overview about the blockchain technology and its characteristics. [R2] is a literature review, where the main contributions about the use of blockchain-based systems in transportation (e.g., smart vehicles security, road traffic management, intelligent transportation systems) are collected and discussed. [R3] and [R4] propose some quite recognized applications about: interconnected smart vehicles and vehicles & forensic applications, respectively. 

[R1] Christidis et al. (2016). Blockchain and Smart Contracts for the Internet of Things, IEEE Access, Vol. 4, pp. 2292-2303. 

[R2] Astarita et al. (2020). A Review of Blockchain-Based Systems in Transportation, Information, Vol. 11, No. 1.

[R3] Dorri et al. (2017). BlockChain: A Distributed Solution to Automotive Security and Privacy, IEEE Communications Magazine, Vol. 55, pp. 119-125.

[R4] Cebe et al. (2018). Block4forensic: An integrated lightweight blockchain framework for forensics applications of connected vehicles, IEEE Communications Magazine, Vol. 56, pp. 50-57. 

Author Response

Reviewer

We thank the reviewer for this insightful comment. In responses, the text of the manuscripts was modified in the following manner( page 9, line 15-20).  And I appreciated the constructive criticisms of distribute of vehicle data by Did with block-chain. We realized that the initial text may have been unclear, So we changed the section 4,5.  We checked papers what you suggested,  “Blockchain and Smart Contracts for the Internet of Things”, “BlockChain: A Distributed Solution to Automotive Security and Privacy”, “ An integrated lightweight blockchain framework for forensics applications of connected vehicles”

The entire manuscript has been carefully edited. As a result, the clarity and readability of the manuscript have been improved.

We look forward to hearing from you in due time regarding our submission and are happy to respond to any further questions and comments you may have.

Round 3

Reviewer 2 Report

The quality of the manuscript has been improved. 

I have only some further suggestions:

1.First of all, as I suggested in the previous report, I think that also the paper [R2] should be cited in Section 4 because it is a point of reference in terms of the use of blockchain-based systems in transportation (e.g., smart vehicles, security, road traffic management, intelligent transportation systems).

[R2] Astarita et al. (2020). A Review of Blockchain-Based Systems in Transportation, Information, Vol. 11, No. 1.

2. Please, at the end of Section 1, you should present the rest of the paper, in order to make aware the reader about the contents. I mean: "Section 2 presents... . In Section 3, we do this... . In Section 5, some conclusions are outlined. etc."

3. Please, write only few lines, specifying your contribution, by making a brief comparison with the existing literature. It is very important to improve the overall quality of the manuscript, and obviously you could use paper [R2] for more information about the state of the art. 

Author Response

I am pleased to resubmit for publication the revise version of “ Manuscript applsci-1057555 : Study on DID Application Methods for Blockchain Based Traffic Forensic Data“.

I appreciated the constructive criticisms of the Associate Editor and the reviewers. I have addressed each of their concerns as yellow outlined in manuscript.

We thank the reviewer for this insightful comment. In responses, the text of the manuscripts was modified in the following manuscript. 

And I appreciated the constructive criticisms of distribute of vehicle data with block-chain in transportation. 

We check papers, A Review of " Blockchain-Based Systems in Transportation.”

The entire manuscript has been carefully edited. As a result, the clarity and readability of the manuscript have been improved.

We look forward to hearing from you in due time regarding our submission and are happy to respond to any further questions and comments you may have

1. In section 4, we quoted “A Review of Blockchain-based Systems of Transportation”. This is referenced from the point of view of using a blockchain-based system for data produced in transport.

2.     At the end of section 1, the direction of the thesis was presented so that the reader could be informed of the content, clearly explaining what the thesis claimed.

3.     Existing thesis mainly emphasized data distribution and integrity through blockchain, but our paper emphasizes the distribution of vehicle accident data through self-sovereign identity technology.

 Since the accident situation of an autonomous vehicle in Level 4 is important, the distribution of metadata at the time of the accident based on blockchain will be very helpful in solving vehicle accidents.

The entire manuscript has been carefully edited. As a result, the clarity and readability of the manuscript have been improved.

We look forward to hearing from you in due time regarding our submission and are happy to respond to any further questions and comments you may have.
